# Engineering Chimeras by Fusing Plant Receptor-like Kinase EMS1 and BRI1 Reveals the Two Receptors’ Structural Specificity and Molecular Mechanisms

**DOI:** 10.3390/ijms23042155

**Published:** 2022-02-15

**Authors:** Qunwei Bai, Chenxi Li, Lei Wu, Huan Liu, Hongyan Ren, Guishuang Li, Qiuling Wang, Guang Wu, Bowen Zheng

**Affiliations:** 1College of Life Sciences, Shaanxi Normal University, Xi’an 710119, China; baiqw132072@163.com (Q.B.); lichenxi_91@163.com (C.L.); wulei199206@163.com (L.W.); 15232019714@163.com (H.L.); hong@snnu.edu.cn (H.R.); guishuangli@snnu.edu.cn (G.L.); gwu3@snnu.edu.cn (G.W.); 2Key Laboratory of Cell Activities and Stress Adaptations, Ministry of Education, School of Life Sciences, Lanzhou University, Lanzhou 730000, China; wangqiuling198@163.com

**Keywords:** brassinosteriods, BRI1, EMS1, receptor function

## Abstract

Brassinosteriods (BRs) are plant hormones essential for plant growth and development. The receptor-like kinase (RLK) BRI1 perceives BRs to initiate a well-known transduction pathway which finally activate the transcription factors BZR1/BES1 specifically regulating BR-mediated gene expression. The RLK EMS1 governs tapetum formation via the same signaling pathway shared with BRI1. BRI1 and EMS1 have a common signal output, but the gene structural specificity and the molecular response remain unclear. In this study, we identified that the transmembrane (TM), intracellular juxtamembrane (iJM), kinase, and leucin-rich repeats 1-13 (LRR1-13) domains of EMS1 could replace the corresponding BRI1 domain to maintain the BR receptor function, whereas the extracellular juxtamembrane (eJM) and LRR1-14 domains could not, indicating that the LRR14-EJM domain conferred functional specificity to BRI1. We compared the kinase domains of EMS1 and BRI1, and found that EMS1’s kinase activity was weaker than BRI1’s. Further investigation of the specific phosphorylation sites in BRI1 and EMS1 revealed that the Y1052 site in the kinase domain was essential for the BRI1 biological function, but the corresponding site in EMS1 showed no effect on the biological function of EMS1, suggesting a site regulation difference in the two receptors. Furthermore, we showed that EMS1 shared the substrate BSKs with BRI1. Our study provides insight into the structural specificity and molecular mechanism of BRI1 and EMS1, as well as the origin and divergence of BR receptors.

## 1. Introduction

Plants have evolved a large number of cell-surface receptor proteins to sense and respond to developmental and environmental cues. Many of these receptors are leucine-rich repeat receptor-like kinases (LRR-RLKs) with (i) an extracellular LRRs domain perceiving the cognate ligands, (ii) a transmembrane domain, and (iii) a cytoplasmic kinase domain activating downstream signaling [1,2]. In *Arabidopsis thaliana*, the BRASSINOSTEROID INSENSITIVE 1 (BRI1) LRR-RLK is found to perceive the plant growth hormones brassinosteroids (BRs); when BRI1 binds with BRs, an intracellular signaling cascade can be initiated, inducing the accumulation of non-phosphorylated forms of two transcription factors, BRI1 EMS SUPPRESSOR 1 (BES1) and BRASSINAZOLE RESISTANT 1 (BZR1), in the nucleus [3,4,5,6,7,8]. Activated BES1 and BZR1 can then mediate the expression of thousands of downstream responsive genes [9,10,11]. The EXCESS MICROSPOROCYTES1 (EMS1) LRR-RLK and its small protein ligand TAPETUM DETERMINANT1 (TPD1) are required for determining the fate of tapetal cells via a signaling transduction pathway shared with BRI1 [12,13,14,15,16]. To function properly, both BRI1 and EMS1 need the co-receptors from the SOMATIC EMBRYOGENESIS RECEPTOR-LIKE KINASE (SERK) subfamily of LRR-RLKs [17,18,19]. Upon the ligand perception, BRI1 and EMS1 with SERKs lead to the trans-phosphorylation between the cytoplasmic kinase domains of receptors and SERK co-receptors, and to the subsequent activation of the cytoplasmic signaling events [17,20].

BRI1 is a well-studied LRR-RLK that functions in regulating seed germination, skotomorphogenesis, flowering, fertility, leaf senescence, vascular differentiation, and stress tolerance [3,21,22]. The BRI1 extracellular region consists of 25 LRRs, interrupted by a 70-amino acid (island domin, ID) chain between the 21st and 22nd LRRs [23,24,25]. The BRs bind to a hydrophobic groove between the ID and the concave side of the LRRs, as well as to the conformationally rearranged ID, which facilitates BRI1 heteromerization with BAK1 [18,19,23,24]. The identified alleles with mutations either in the ID or at the ID-LRR interface have revealed the significance of this domain [3,21,26,27]. The cytoplasmic domain of BRI1 can be subdivided into a juxtamembrane (JM) region, a serine/threonine/tyrosine kinase domain (KD), and a C-terminal (CT) domain [25]. Among all plant RLKs, the KD domain is conserved and has a common activation mechanism, whereas the JM domain and CT domain exhibit more diversity [28,29].

Previous research has identified many phosphorylation sites of BRI1 by liquid chromatography-tandem mass spectrometry (LC-MS/MS), and eliminating phosphorylation at these residues alters the kinase activity of BRI1 in various ways [30,31,32,33]. Several sites have been confirmed as indispensable for kinase activity and plant growth. For example, mutants affecting phosphorylation of Tyr-1052 abolished BRI1 kinase activity and prevented rescue of the dwarfism of *bri1-5* [32,34]. In Arabidopsis, three homologs were identified for BRI1, BRI1-LIKE 1 (BRL1), BRI1-LIKE 2(BRL2), and BRI1-LIKE 3 (BRL3) [35]. Upon the expression of these homologs in *bri1* mutants, their phenotypes were rescued by BRL1 and BRL3, but not BRL2, indicating that BRL1 and BRL3, but not BRL2, can interact with BL [35,36].

EMS1 protein shares some common features with BRI1, including an extracellular domain with 30 LRRs, a transmembrane domain, and a cytoplasmic protein kinase domain [16]. Our previous study revealed that the KD of EMS1 and BRI1 are interchangeable in *Arabidopsis* and able to activate the same signaling pathway, suggesting the KD molecular functions of EMS1 and BRI1 are the same [14]. Furthermore, the EMS1-TPD1 pair is present in all land plants, whereas BRLs are apparently absent from non-vascular plants [14,37]. Thus, EMS1 and BRLs appear to originate in a common ancestor that probably transmits signaling through the BZR1/BES1. According to our hypothesis, the BRI1-BRL family evolved to combine smaller and more diffuse molecules, resulting in a broader expression pattern, whereas EMS1 senses short-range peptide signals that are limited to specific tissues [38]. These receptors would have subsequently undergone duplication and divergence events that have given rise to the BRL family of angiosperms. The most critical molecular event in the origin of BRL is that the extracellular domain acquired the capacity to bind BRs [38,39]. As an important piece of evidence, the engineering of a more ancient EMS1 to obtain a functional receptor for BRs may give us valuable insights into understanding the origin of the BR receptor.

Here, we report that the LRR14-EJM domain confers the functional specificity to BRI1, with at least three key subdomains included: LRR13-LRR14, ID-LRR and ECD. We also show the kinase activity of EMS1 is weaker than that of BRI1, and identify a site specificity between EMS1 and BRI1. Furthermore, we reveal that EMS1 shares the substrate BSKs with BRI1, which provides a biochemical basis for similar molecular functions of the two receptors. Thus, the specificity of EMS1 and BRI1 underpins the divergence of the respective receptors, and the key motifs essential for the origin of BR receptors.

## 2. Results

### 2.1. Transmembrane, Intracellular Juxtamembrane, and Kinase Domains, but Not the Extracellular Jux-Tamembrane Domain, of EMS1 Can Completely Replace the Corresponding BRI1 Domain

Our previous study uncovered that the intracellular domains of EMS1 can substitute the intracellular domains of BRI1 to maintain the typical receptor function [14]. However, it remained unclear whether the subdomain outside of the EMS1 kinase domain could replace the corresponding BRI1 domain. In order to seek the structural subunits in EMS1 that can replace BRI1, we divided the functional domains of BRI1 and EMS1 based on the mentioned structural features. Using the domain swapping method, we generated several chimeric receptors by fusing BRI1 and EMS1, including BE(KD), BE(TM), BE(eJM), and BE(ID) (Figure 1A). BE(KD), consisting of BRI1 extracellular domain, transmembrane domain, and intracellular juxtamembrane domain fused to the kinase domain of EMS1. BE(TM) with BRI1 extracellular and transmembrane domains fused to the intracellular juxtamembrane and kinase domain of EMS1. BE(eJM), the chimeras with BE(TM) in the absence of BRI1 extracellular juxtamembrane domain. BE(ID), the chimeras with BE(eJM) in the absence of BRI1 ID adjacent LRR domain (Figure 1A). We transformed these chimeric receptors into a weak BRI1 mutant *bri1-301*. The results showed that BE(KD) and BE(TM) were able to completely rescue the phenotype of *bri1-301* (Figure 1B,C), and that the BE(eJM) partially restored *bri1-301* phenotype, whereas the BE(ID) failed to repress the phenotype of *bri1-301* (Figure 1B,C). The hypocotyl length of the seedling in the dark can be used as an indicator of the strength of the endogenous BR signal [3]. We observed that BE(KD) and BE(TM) completely recovered the defective hypocotyl elongation of seedling grown in the dark (Figure 1B–E). The hypocotyl was longer in the *BE(eJM)* transgenic plants than *bri1-301*, but not as long as in the *BE(KD)*, *BE(TM)*, and wild-type, which was consistent with the rosette leaves phenotype (Figure 1B–E). To further assess the restoration of the chimeric gene in *bri1-301*, we used semi-quantitative PCR to analyze the expression of BR biosynthetic genes *CPD* and *DWF4* or BR catabolic gene *BAS1* in the transgenic plants [40]. We found that the expression of *CPD* and *DWF4* was dramatically down-regulated and that of *BAS1* was up-regulated in *BE(KD)*/*bri1-301* and *BE(TM)*/*bri1-301* compared with *bri1-301* mutant. The transcription levels of *CPD*, *DWF4*, and *BAS1* in *BE(eJM)*/*bri1-301* were slightly regulated in comparison with *bri1-301*, whereas *BE(ID)* and *EMS1* did not have an effect on gene expression (Appendix A).

To test whether the function of chimeric receptors depend on BRs, we applied BL (brassinolide, the most active BR) to transgenic plants. We observed that *BE(KD)*/*bri1-301*, *BE(TM)*/*bri1-301*, and *BE(eJM)*/*bri1-301* were hypersensitive to BL, but *BE(ID)* and *EMS1* were insensitive, indicating that BE(KD), BE(TM), and BE(eJM) have the BR receptor functions, whereas BE(ID) and EMS1 do not have such functions (Appendix A). We further examined the phosphorylation status of BES1 after exogenous BL treatment [7]. In *BE(KD)*, *BE(TM)* and *BE(eJM)* transgenic plants, we detected the accumulation of dephosphorylated BES1 in contrast to *bri1-301* mutant. However, in *BE(ID)* and *EMS1* transgenic plants, the dephosphorylated BES1 was similar to that observed in *bri1-301* plant (Figure 1F). Interestingly, the proportion of dephosphorylated BES1 was lower in the *BE(eJM)* transgenic plants than the *BE(TM)* and *BE(KD)* plants (Figure 1F). Taken together, these results suggested that the KD and TM domains in EMS1 could replace these domains in BRI1 to maintain the BR receptor function, whereas the eJM domain of EMS1 diminished the BR receptor function. The ID adjacent LRR was crucial for the BR receptor functions and could not be replaced by EMS1.

### 2.2. The LRR14-eJM Domain Confers the Functional Divergence between BRI1 and EMS1

BR receptors rely on the island domain (ID) that is located in the extracellular domain to recognize and bind BR and transmit the BR signal [4]. The *BE(ID)* chimeric receptor could not restore the mutant, suggesting that BR driving BR receptor function did not depend only on the island domain but also on the ID adjacent LRR domain (Figure 1). In order to explore the functionally specific domains in BRI1, we fused different lengths of EMS1 LRR sequence fragments with the LRR-ID-ICD domain of BRI1 to generate the chimeric constructs (Figure 2A). According to the number of intercepted LRRs, these constructs were named EB(LRR7), EB(LRR13), EB(LRR14), and EB(LRR17) (Figure 2A). After these chimeras were transformed into *bri1-301*, we found that *EB(LRR7)* and *EB(LRR13)* repressed the *bri1-301* rosette leaves and root elongation, whereas *EB(LRR14)* and *EB(LRR17)* did not have such an effect (Figure 2A and Appendix A). Consistent with the observed phenotype, the expression level of *CPD*, *DWF4*, and *BAS1* in *EB(LRR7)* and *EB(LRR13)* transgenic plants was similar to that in Col-0, but not in *bri1-301* (Appendix A). In order to verify the function of chimeric receptors dependent on BR, we measured the root length of the transgenic plants after applying a high concentration of exogenous BR. The results showed that *EB(LRR7)*/*bri1-301* and *EB(LRR13)*/*bri1-301* were sensitive to BL, whereas *EB(LRR14)*/*bri1-301* and *EB(LRR17)*/*bri1-301* did not respond to BL (Appendix A). These results indicated that the chimeric receptors EB (LRR7) and EB (LRR13) maintained the BR receptor function, but EB(LRR14) and EB(LRR17) did not. In the first part of this study, we demonstrated the TM and ICD domains in EMS1 could replace the corresponding BRI1 domain to maintain the BR receptor function completely. We then generated the chimeric EBE(LRR13) comprising the BRI1 LRR14-eJM domain fused to the other domain of EMS1 (Figure 2A). The genetic experiment and the BR treatment experiment confirmed that EBE (LRR13) had the complete BR receptor function. (Figure 2B–D). Collectively, we have proved that the BR receptor function can be obtained by modifying the extracellular LRR14-eJM segment of EMS1 fused with BRI1.

The first three LRRs of EMS1 were sufficient for interacting with TPD1 to activate EMS1-TPD1 signaling [41]. We then tested whether the receptors EB(LRR7) and EB(LRR13) that contain the first three LRRs of EMS1 have a dual function of binding to both TPD1 and BR. Our previous study showed that the co-expression of EMS1&TPD1 repressed the *bri1* mutant, but made it insensitive to Brz (Brassinozole, BR synthesis inhibitors), which can be used as a strong indicator to test the TPD receptor function [14]. In order to verify this hypothesis, we applied Brz to the *EB(LRR7)&TPD1*/*bri1-301*, *EB(LRR13)&TPD1*/*bri1-301*, and *EMS1&TPD1*/*bri1-301* co-expression transgenic plants, with *EMS1&TPD1*/*bri1-301* used as a positive control (Figure 2E,F). We found that *EB(LRR7)&TPD1*/*bri1-301* and *EB(LRR13)&TPD1*/*bri1-301* were sensitive to Brz. When treated with Brz, the hypocotyl elongation of *EB(LRR7)&TPD1*/*bri1-301* and *EB(LRR13)&TPD1*/*bri1-301* was suppressed (Figure 2E,F). These results suggest that the 1-13 LRR domain of EMS1 is not sufficient for the TPD1 receptor function when fused with BRI1.

### 2.3. The Kinase Activity of EMS1 Is Weaker than That of BRI1

The intracellular domains (ICD) of BRI1 and EMS1 are functionally interchangeable, indicating similar molecular functions [14]. When comparing the ICD sequence of BRI1 and EMS1, there was still a large number of distinct sites, suggesting different kinase activities of EMS1 and BRI1 (Appendix A). To investigate the distinction between the intracellular domains of EMS1 and BRI1, we constructed a chimeric vector in which the extracellular domain of EMS1 was fused with the intracellular domain of BRI1 (Figure 3A). *EMS1* and *EMS1-BRI1* were co-transformed into *bri1-116* with *TPD1*, respectively. Similar to *EMS1&TPD1*, the *EMS1-BRI1&TPD1* restored *bri1-116*, but the rosette leaves and roots of *EMS1-BRI1&TPD1*/*bri1-116* transgenic plants showed an extremely distorted phenotype akin to those usually associated with a robust BR signal output. (Figure 3B,C and Appendix A).

We further assessed the BR-associated genes *CPD*, *DWF4*, and *BAS1*, and found that the down-regulation of *CPD* and *DWF4* and the up-regulation of *BAS1* were more severe in the *EMS1-BRI1&TPD1*/*bri1-116* transgenic plants than *EMS1&TPD1*/*bri1-116* (Appendix A). We also transformed *EMS1* and *EMS1-BRI1* into *bri1-301* and found that the *EMS1-BRI1*/*bri1-301* plants showed the phenotype of rosette leaf curl in contrast with *EMS1*/*bri1-301* (Figure 3D and Appendix A). Similar to co-expressing the BR-related genes in the *bri1-116* background, these genes were regulated to a greater extent in *EMS1-BRI1*/*bri1-301* than in *EMS1*/*bri1-301* (Appendix A). These results indicated that the EMS1-BRI1 showed stronger biological activity than EMS1. Both EMS1 and BRI1 belong to RLK, and their molecular functions depend on the auto-phosphorylation and trans-phosphorylation activities of the intracellular kinase domain [17,20,33]. The weaker biological activity of EMS1 may indicate its weaker auto-phosphorylation activity compared with BRI1. Thus, we measured the auto-phosphorylation of BRI1 and EMS1 in vitro; the results show that the autophosphorylation was stronger in the kinase domain of BRI1 than in EMS1 (Figure 3E). In summary, we revealed that, despite EMS1 and BRI1 sharing a common signal output, BRI1 had a stronger kinase activity than EMS1.

### 2.4. The Tyrosine (Tyr) Residues Y1052 Is Essential for BRI1 but the Corresponding Residue Y1085 Is Not Essential for EMS1

BRI1 acts as a dual-specificity kinase that is capable of phosphorylation on both serine/threonine (Ser/Thr) and tyrosine (Tyr) residues [26,30,32,33]. There are many important tyrosine phosphorylation sites in the kinase domain of BRI1, which are essential for plant growth and development [32,34]. A previous study showed that Ser and Thr phosphorylation sites have an effect on EMS1 auto-phosphorylation [17]. We aligned the sequences of EMS1 and BRI1 kinase domains. It is found that, at some tyrosine sites, BRI1 and EMS1 are the same, such as Y956, Y1052, and Y1057 in BRI1 being equivalent to Y990, Y1085, and Y1090 in EMS1, but there were differences at other sites, such as Y1072 in BRI1 and F1105 in EMS1 (Appendix A). In order to test the effect of tyrosine (Tyr) residues on the auto-phosphorylation of EMS1 and BRI1, we performed site-directed mutagenesis of these Tyr to Phe and Phe to Tyr. Y956F, Y1052F, Y1057F, and Y1072F in BRI1-CD, as well as Y990F, Y1085F, Y1090F, and F1105Y in EMS1, were generated to obtain recombinant proteins. Using pThr antibody to detect the phosphorylation level of BRI1 and EMS1, we observed that Y956F, Y1052F, and Y1057F strongly inhibited auto-phosphorylation of BRI1, but Y1072F did not affect the process (Figure 4A). For EMS1, the auto-phosphorylation of EMS1-Y990F-CD, EMS1-Y1085F-CD, and EMS1-F1105Y-CD was significantly decreased and the impact of Y1090F was moderate (Figure 4B). Phosphorylation of specific Tyr residues in LRR-RLK is vital for kinase activation and signal transduction. To determine whether these sites are required for the biological function of BRI1 and EMS1 in vivo, we constructed the site-directed mutants in the full-length BRI1 and the chimeric BRI1-EMS1 (Appendix A). Transforming these constructs into *bri1-116*, *BRI1(Y956F)*, *BRI1(Y1057F)*, *BRI1(Y1072F)*, *BRI1-EMS1(Y990F)*, *BRI1-EMS1(Y1090F)*, and *BRI1-EMS1(F1105Y)* repressed the phenotype of *bri1-116*; *BRI1-EMS1(Y1085F)* partially recued *bri1-116*, but *BRI1(Y1052F)* did not restore the phenotype of *bri1-116* (Figure 4C). These results indicate that Y1052 is essential for the biological function of BRI1 but not BRI1-EMS1.

Y956 and Y1057 regulate the phosphorylation of BRI1 in vitro but do not affect the BRI1 biological function in vivo (Figure 4A,C). In order to precisely evaluate the effects of Y956, Y1057, and Y1072 on the biological functions of BRI1, we constructed double site-directed mutations and then transferred them into *bri1-116*. By analyzing the phenotype of transgenic plants, we observed that *BRI1*(*Y956F Y1072F*) and *BRI1*(*Y1057F Y1072F*) almost rescued the rosette leaves and hypocotyl elongation of *bri1-116*, but that *BRI1*(*Y956F Y1057F*) could not (Appendix A). The transcription levels of BR-related genes *CPD*, *DWF4*, and *BAS1* in *BRI1*(*Y956F Y1072F*)/*bri1-116* and *BRI1*(*Y1057F Y1072F*) /*bri1-116* were comparable to Col-0, and in *BRI1*(*Y956F Y1057F*)/*bri1-116* they were similar to *bri1-116* (Appendix A). These findings were consistent with the phenotype of transgenic plants. An in vitro kinase assay also showed that the phosphorylation levels were undetectable in BRI1(Y956F Y1057F)-CD and were abolished in BRI1(Y956F Y1072F)-CD and BRI1(Y1057F Y1072F) (Appendix A). These results indicated that Y956 and Y1057 were not necessary for the function of BRI1; the mutated BRI1 had a reduced phosphorylation level but still retained the biological function.

Although EMS1 Y1085F significantly decreased the phosphorylation activity of EMS1, the BRI1-EMS1 (Y1085F) still partially restored *bri1-116*, implying that Y1085 was not essential for the EMS1 function (Figure 4B,C). To confirm this claim, we developed the full length, site-directed mutation EMS1 Y1085F and transferred it into the ems1 mutant, finding that EMS1(Y1085F) could still produce fertile pollen (Figure 4D). The co-expressed *EMS1&TPD1* and *EMS1(Y1085F)&TPD1* restored the hypocotyls of the transgenic *bri1-116* plants, whereas the hypocotyl elongation of *EMS1(Y1085F)&TPD1*/*bri1-116* was partially inhibited compared with *EMS1&TPD1*/*bri1-116* (Figure 4E,F). These results indicated that, in contrast to BRI1, for which Y1052 was essential, the Y1085 was not required for the EMS1 function.

### 2.5. EMS1 and BRI1 Share the Substrate BSKs

The downstream components of BRI1 have been identified previously. However, the direct downstream substrates of EMS1 are not clear. The similar molecular functions of EMS1 and BRI1 strongly indicate that they should share the common downstream substrates. BSK1 and BSK3 belong to a receptor-like cytoplasmic kinase subfamily RLCK-XII; previously, they were proven to be direct substrates of BRI1 [42,43]. To test whether BSK1 and BSK3 are the substrates of EMS1, we performed the pull-down assay by obtaining GST-BRI1-CD, GST-EMS1-CD, BSK1-His, and BSK3-His recombinant proteins. The recognized interactions between BRI1, BSK1, and BSK3 were performed as the positive control in this experiment [42,43]. The result showed that, similarly to BRI1, the BSK1 and BSK3 interacted with EMS1 directly (Figure 5A and Appendix A). This result was also confirmed by the bimolecular fluorescence complementation (BiFC) and coimmunoprecipitation (Co-IP) analyses (Figure 5B,C and Appendix A). Epidermal cells of *Nicotiana benthamiana* leaves co-expressing EMS1 fused to the N-terminal half of YFP (nYFP) and BSK1, or BSK3 fused to the C-terminal half of YFP (cYFP), showed strong fluorescence signals. By contrast, co-expressing EMS1-nYFP with cYFP did not generate the fluorescence signals (Figure 5B). We performed the Co-IP assay by expressing *BSK1-FLAG*, *BSK3-FLAG*, and *EMS1-GFP* as negative controls, and co-expressing *BSK1-FLAG* or *BSK3-FLAG* with *EMS1-FLAG* in Col-0. The EMS1 protein was immunoprecipitated by anti-GFP antibodies only in transgenic plants expressing *EMS1-GFP*&*BSK1-FLAG* and *EMS1-GFP*&*BSK3-FLAG* (Figure 5C and Appendix A). These results indicated that BSK1 and BSK3 could interact with EMS1.

The interaction results suggested that EMS1 might directly phosphorylate BSKs to activate EMS1 signaling. To test the ability of EMS1 to phosphorylate BSK1 and BSK3, BSKs were expressed with a His tag in *E. coli* and EMS1 kinase domain was fused to maltose binding protein (MBP) and GST. Because of the similar size of GST-EMS1-CD and BSK1-His, the EMS1-CD-MBP was blended with BSK1-His (and GST-EMS1-CD with BSK3-His) to perform the kinase assay. The results indicated that EMS1 phosphorylated BSK1 and BSK3 (Figure 5D and Appendix A).

## 3. Discussion

Receptor-like kinases (RLKs) are crucial for plants to sense constantly changing environments and then regulate their various growth and development processes [44,45,46]. EMS1 and BRI1 regulate different biological processes through similar signaling events [14,15]. The domain swap assay confirmed that EMS1 and BRI1 could be interchanged and still maintain their biological functions, which suggested that the main differentiation fragment of EMS1 and BRI1 should be associated with the extracellular domain [14]. The evolutionary analysis of terrestrial plants indicates that EMS1 is present in all land plants, and BRLs that can bind to BR exist in seed plants [37,39]. How did BR receptors evolve? The study of the structural differences between EMS1 and BRI1 provides a good basis for understanding this process. Therefore, the essential question of our research was to attempt to engineer EMS1 from a small-peptide TPD1 receptor to a plant hormone BR receptor. We discovered that the kinase domain (KD), intracellular juxtamembrane domain (iJM), transmembrane domain (TM), and the 1–13 LRR domain of EMS1 could replace the corresponding BRI1 domains, suggesting that the key domain of BR receptor origin comprised approximately 400 amino acids, i.e., the LRR14-eJM segment (Figure 1 and Figure 2, Appendix A). More detailed analysis suggested that this segment included at least three core motifs: LRR14, ID-LRR22, and eJM. The ID-LRR22 motif that is responsible for the perception of BRs has been thoroughly studied [4]. Our present work showed that the LRR14 and eJM regions of EMS1 could not fully replace the corresponding segments of BRI1, indicating that the origin of BRI1 involved the differentiation of these two key motifs as well as the ID-LRR22 motif (Figure 1 and Figure 2, Appendix A). In which, the eJM motif of BRI1 can be replaced by EMS1 at least partially, but the LRR14 motif could not be replaced at all, indicating that LRR14 motif was more essential for the BRI1 function (Figure 1 and Figure 2, Appendix A). There is still scant knowledge of how the two motifs LRR14 and eJM affect the BR receptor functions; the relevant mechanism might involve the stabilization of the receptor conformation or a mode of signal transmission.

The EMS1 and BRI1 signaling pathways regulate different biological processes and this distinction is mainly caused by the difference in expression pattern. To shed more light on the function divergence of EMS1 and BRI1, we used their own promoters to drive GUS to analyze the expression in different tissues. The results showed that *BRI1*, *EMS1*, and *TPD1* were expressed broadly, and BRI1 was expressed more highly than the other two in younger tissues (Figure 6). The results of real-time quantitative PCR in specific tissues confirmed that EMS1 was expressed in siliques and inflorescence, which was consistent with its biological function. The TPD1, ligand of EMS1, was found in roots, seedlings, leaves, siliques, and inflorescence (Appendix A). At present, the significance of the EMS1-TPD1 signaling pathway in tissues other than in flower organs has yet to be ascertained.

Receptor-like kinases (RLK) are the largest receptor family in plants. With the evolution of terrestrial plants, the functions of RLKs have expanded and diversified, giving plants more sensitive signal perception and transduction [1,37]. How did the complex and diverse receptor kinases evolve? Our investigation of EMS1 and BRI1 suggests that the evolutionary pattern of different receptors may involve a combination of building blocks. The main distinction between EMS1 and BRI1 is the ligand-binding fragment that is in an extracellular domain. Obviously, EMS1 contains the BR-binding sequence motif in the extracellular domain, which can be taken as the origin of BR receptors. Such evolution of domain configuration can occur in both extracellular and intracellular domains (Figure 7). GSO1 differs from BRI1 and EMS1 in intracellular kinase function [8,16,47]. Recent research has shown that GSO1 kinase gains the function of a BR receptor after acquiring two intracellular motifs, Appendix A [48]. The three-part structure of receptor-like kinases (intracellular, transmembrane, and extracellular) makes it easier to build this module. One piece of evidence supporting this speculation is that RLKs can produce new functions after forming a chimeric protein. Engineered Rid-BRI1 and RiD-FLS2 could trigger BR-independent and flagellin-22-mediated responses after rapamycin treatment, respectively [49]. The chimeric receptor BRI1-XA21 initiates plant defense responses in rice cells treated with BRs [50]. Upon stimulation with the elf18, the EFR-WAK1 activates the defense responses [51]. The BIR3-BRI1 chimeric receptor can support continuous BR signaling activation [52]. These domain swap strategy can be regarded as a process of receptor functional differentiation and origin that imitates the natural state. Our previous investigation has shown that the kinase functions of EMS1 and BRI1 are similar [14]. In the present study, we have revealed the differences in the kinase domain. Although they have the same downstream activation components, BRI1 has a stronger kinase activity than EMS1. When forming the chimeric receptor, EMS1-BRI1 showed a stronger phenotype, and further testing confirmed that the autophosphorylation activity of the BRI1 kinase domain was superior (Figure 3, Appendix A). BR is the most effective among plant hormones. One possibility is that the kinase activity of the BR receptor has been enhanced continuously during the evolution of BR receptors, which may be an economical way of allowing the receptor to produce a range of consequences with a few ligands.

We noticed that EMS1 F1105, a natural differential site in the kinase region, is different with BRI1. Further analysis of the four tyrosine sites revealed the kinase specificity in autophosphorylation sites between EMS1 and BRI1. In the site-directed mutation, we discovered that only Y1052 of the four tyrosine sites was indispensable for BRI1 (Figure 4). Further double-sites mutations confirmed that the other two sites (Y956F and Y1057F) decreased the kinase activity, whereas the Y1072F had nearly no impact on the kinase activity and biological activity of BRI1 (Appendix A). This conflicts with the published literature that indicated these four sites completely lost kinase activity and biological function after simulated mutations [32,34]. We suspect that the current literature has been compromised by the use of the weak mutant *bri1-5*, which leads to a deceptive phenotype. To avoid this situation, we utilized the *BRI1* null mutant *bri1-116*. The evaluation of the four corresponding sites in EMS1 confirmed that Y1072 should not affect biological function. The corresponding site in the BRI1-EMS1 chimeric receptor is a natural non-phosphorylation site, but BRI1-EMS1 and BRI1-EMS1(F1105Y) mutation still recovered the *bri1-116* phenotype (Figure 4C). This result is consistent with the autophosphorylation and biological analysis of BRI1(Y1072) (Figure 4A). We proved that another site in BRI1 (Y1052) was essential, but that the corresponding site in EMS1 (Y1085) only weakened the receptor function (Figure 4). EMS1 (Y1085F) and BRI1-EMS1 (Y1085F) still retained most of their biological function (Figure 4C–F). This suggests that even though EMS1 and BRI1 are the same in downstream core transduction components, there are significant differences in the mechanism regulating the kinase activity. We explain that this regulatory distinction at a specific site may fine-tune the signal output of the receptors.

EMS1 and BRI1 activate BZR1/BES1. We thus speculated that their direct components should be shared. The protein interaction assays confirmed this hypothesis (Figure 5 and Appendix A). EMS1 and BRI1 share the same downstream substrate BSKs, which is the molecular basis of their identical signal output (Figure 7). However, this sharing of transduction components does not mean that the signal properties produced by the EMS1 and BRI1 are exactly alike. For example, BKI1 as one of the relevant components of BRI1 binds to the C-terminal domain of BRI1 to negatively regulate the BR signaling, In fact, EMS1 does not interact with BKI1 because of the absence of the C-terminal domain in EMS1 [53,54], suggesting that BKI1 is a specific component of BRI1. In addition, EMS1 can activate the BR signal without binding to BKI1, indicating that BKI1 is not a required component of the BR signaling pathway. By contrast, βCa is a substrate of EMS1 and does not interact with BRI1, suggesting that βCa is a specific component of EMS1 [55]. Somatic embryogenesis receptor-like kinases (SERKs) function as co-receptors for RLKs involved in several plant-signaling pathways [56]. Interestingly, SERK3 (BAK1) can directly interact with BRI1 but not with EMS1 [17,18,19]. It has been shown that the difference in the extracellular structure of EMS1 and BRI1 is responsible for EMS1 not recognizing BAK1 [17]. It remains unclear how different signal components of EMS1 and BRI1 affect the signal output. Obviously, EMS1 and BRI1 differ not only in signal strength but also in signal properties.

In the present study, we compared the differences in gene structure and differentiation between EMS1 and BRI1 via developing chimeric receptors and revealed the functionally specific core motifs of the BR receptor BRI1. We also compared the functions of the EMS1 and BRI1 kinase domains (Figure 7). These results are significant for an in-depth understanding of the functions of EMS1 and BRI1 signaling pathways, as well as of generating and differentiating new RLKs functions.

## 4. Methods and Materials

### 4.1. Plant Materials and Growth Conditions

The *Arabidopsis thaliana* Columbia (Col-0) and Landsberg *erecta* (*Ler*) ecotypes were used as wild type control in this study. *bri1-116* and *bri1-301* mutants are in Col-0 background. *ems1* is in the *Ler* background. Seeds were germinated into 1/2 Murashige & Skoog (MS) medium, then transferred into the soil. Plants were grown under long daylight conditions (16-h light/8-h dark cycles). Seedlings for testing BR-sensitive and BR-dependent responses were germinated on 1/2 MS medium with or without BL, or Brz in light or in dark, respectively. The root length and hypocotyl length were measured using ImageJ software.

### 4.2. Constructs and Transgenic Plants Generation

The sequences of *BRI1* and *EMS1* were inserted into *pBRI1::GFP* or *pEMS1::GFP* to complement the *bri1* or *ems1* mutants as indicated in the figures. The cDNA sequences of *BSK1* and *BSK3* were introduced into *pBRI1::FLAG*, respectively. Overlapping PCR was used to generate chimeric and site-directed mutant vectors. We amplified the promoters of *BRI1*, *EMS1*, and *TPD1* into the pCAMBIA2300 vector with GUS. All constructs were transferred into plants via *Agrobacterium* (GV3101)-mediated transformation using the floral dip method. The chimeric and site-directed mutant vectors were transformed into *bri1-301* and *bri1-116*, as indicated in the figures. The *EMS1^Y1085F^* was transformed into *ems1*. The BRI1-GUS, EMS1-GUS and TPD1-GUS plasmids were transformed into Col-0 to investigate the expression pattern. The transformants were screened on 1/2 MS with 50 μg/mL kanamycin or 40 μg/mL hygromycin B. Double transgenic plants were produced by crossing and screening of 1/2 MS with 50 μg/mL kanamycin and 40 μg/mL hygromycin B. The primers used in this study were given in Appendix A.

### 4.3. Protein Extraction and Immunoblot Analysis

For detecting BES1 phosphorylation status, 14-day-old seedlings were treated with or without 1 μM BL for 1 h and then the proteins were extracted using 2 × SDS buffer (100 mM Tris, pH 6.8, 4% [*w*/*v*] SDS, 20% [*v*/*v*] glycerol, 0.2% [*w*/*v*] bromophenol blue, 2% [*v*/*v*] β-Mercaptoethanol). For testing the gene expression, total proteins were obtained from 4-week-old rosette leaves using 2 × SDS buffer. Then, the proteins were separated on SDS-PAGE gel and transferred onto a Nitrocellulose membrane. Anti-GFP antibody (1:1000 dilution, Transgen, HT801) was used to detect GFP fusion proteins and BES1 antibody (1:3000 dilution, kindly provided by Jia Li, Lanzhou University, Lanzhou, China) was used to detect the phosphorylation status of BES1. The equal loading was evaluated by β-tubulin (C66) mAb (1:1000 dilution, Abmart, Shanghai, China, M20005M).

### 4.4. Pollen Staining

The flower buds were fixed in Carnoy’s fluid (alcohol: chloroform: acetic acid = 6:3:1) for 2 h. Individual anthers were dissected and stained with Alexander’s staining solution at 50 °C for 48 h. The anthers were visualized and photographed under a compound microscope.

### 4.5. RT and qRT-PCR Analysis

Total RNA was isolated from the root, seedings, leaves, inflorescence, and siliques using a HiPure Plant RNA Mini Kit (Magen, R4151-02) according to the protocol. First-strand cDNA was synthesized from 2 μg of total RNA using 5× All-In-One RT MasterMix (abm, G490). The qRT-PCR was performed using ChamQ^TM^ SYBR qPCR Master Mix (Vazyme, Q311-00) to detect the transcript levels of genes. For RT-PCR, the cDNA was amplified with specific primers for *CPD*, *DWF4*, *BAS1*, and *ACT2*. The primers used for RT-PCR and qRT-PCR are listed in Appendix A.

### 4.6. Bimolecular Fluorescence Complementation Assay

Protein–protein interactions were performed using bimolecular fluorescence complementation (BiFC) in *N. benthamiana*. *N. benthamiana* was sown directly in soil and grown under long daylight conditions (16-h light/8-h dark cycles). The full length of EMS1 was cloned into pSPYNE-173. The full length of BSK1 and BSK3 was introduced into pSPYCE-M. *A. tumefaciens* strains (GV3101) taking the constructs were used together with the p19 strain and re-suspended in infiltration buffer (0.15 M acetosyringone dilute in DMSO; 0.01 M MES, pH 5.7; 0.01 M MgCl_2_). The mixed solution was incubated at room temperature for 2 h, and then used for direct infiltration of 4-week-old *N. benthamiana* leaves. After 36 h–48 h, the leaf sections were excised and visualized by confocal fluorescence microscopy.

### 4.7. Coimmunoprecipitation

For Co-IP assay, we constructed BSK1-FLAG and BSK3-FLAG vectors with a C-terminal fusion of 3× FLAG tag and a EMS1-GFP vector with GFP tag fused to the C-terminus of EMS1. These vectors were transformed into Col-0 to obtain transgenic plants. The proteins were harvested from Col-0, single transgenic, and double transgenic plants with extracting buffer (10 mM HEPES, pH 7.5; 100 mM NaCl; 1mM EDTA pH 8.0; 10% Glycerol; 0.5% Triton X-100; 1× cocktail). Total Proteins were incubated with anti-Flag M2 affinity gel (sigma, St. Louis, MO, USA, A2220). Anti-Flag M2 affinity gel binds FLAG tags at the C-terminus of BSK1 and BSK3. The immunoprecipitants were separated on SDS-PAGE and analyzed by Western blot with anti-GFP (1:1000 dilution, Transgen, HT801) and anti-FLAG (1:1000 dilution, Abmart, M20008F).

### 4.8. In Vitro Kinase Assays

To obtain proteins for in vitro kinase assay, the cytoplasmic domains of EMS1 (849-1192 AA) were cloned into pGEX-4T-3 and pMAL-c2x vector and to create GST-EMS1-CD and MBP-EMS1-CD, respectively. The cytoplasmic domains of BRI1 (amino acids 814-1196 AA) were amplified into pGEX-4T-3. The full-length cDNA of BSK1 and BSK3 were inserted into the vector pET-28a. Site-direct mutations of BRI1-CD and EMS1-CD were generated using overlapping PCR. Primers for generating all constructs are included in Appendix A. The constructs were transferred into *E. coli* strain BL21 (DE3) and the recombinant fusion proteins were induced with 0.3 mM Isopropyl-β-D-thiogalactopyranoside (IPTG) at 16 °C for 14–16 h. The GST fused protein, His fused, and MBP fused proteins were purified by Glutathione Sepharose 4B (GE Healthcare, Chicago, IL, USA, 17-0756-01), Ni-NTA Agarose (QIAGEN, Hilden, Germany, 151032765), and Amylose Resin (NEB, E8021V), respectively. For kinase assay, recombinant proteins were added to the 20 μL reaction buffer (20 mM Tris-HCl, pH 7.4, 100 mM NaCl, 10 mM MnCl_2_, and 1 mM ATP) at 30 °C for 1 h. The reactions were added at an equal volume of 2 × SDS-PAGE loading buffer and boiled for 5 min at 95–100 °C. The proteins were resolved by SDS-PAGE. For auto-phosphorylation, the amount of recombinant fusion proteins was detected by GST antibody (1:2000 dilution, GE Healthcare, 02902N) and His-Tag antibody (1:2000 dilution, Abmart, M30111) antibody. For trans-phosphorylation, the protein is stained by Coomassie Bright Blue (CBB). The phosphorylation level was detected by immunoblotting with Phospho-Threonine Antibody (1:2000 dilution, CST, 9381).

### 4.9. Pull-Down

Add BSK1-His or BSK3-His into GST-BRI1-CD, GST-EMS1-CD, and GST proteins incubated with 20 µL Glutathione Sepharose 4B beads (GE Healthcare, 17-0756-01). After 4–6 h gentle rocking at 4 °C, the proteins were washed 6–8 times with 50 volumes of His lysis buffer, stopped by adding an equal volume of 2× SDS buffer, and then detected with His antibody and GST antibody. The reaction of GST with BSK1-His or BSK3-His is regarded as the negative control.

### 4.10. GUS Staining

Plant tissues were fixed for 1 h in acetone at −20 °C. After washed in washing buffer [0.1 M phosphate (pH 7.0), 10 mM EDTA, and 2 mM K_4_Fe(CN)_6_] twice for 5 min, the materials were transferred into the GUS staining buffer [0.1 M phosphate (pH 7.0), 10 mM EDTA, 0.5 mM K_4_Fe(CN)_6_, 0.1% (*v*/*v*) TritonX-100 and 1 mg/mL X-GLUC)] and incubated for 3–5 h at 37 °C. Then, they were detained in 75% ethanol several times. The plant materials were observed under a microscope.

### 4.11. Statistical Analysis

Statistical analysis was performed using two-way ANOVA with Sidak’s test, one-way ANOVA with a Tukey’s test, and two-tailed *t*-tests, as implemented in GraphPad Prism 8.0. (GraphPad Software, http://www.graphpad.com (accessed on 12 February 2022)).

## Figures and Tables

**Figure 1 ijms-23-02155-f001:**
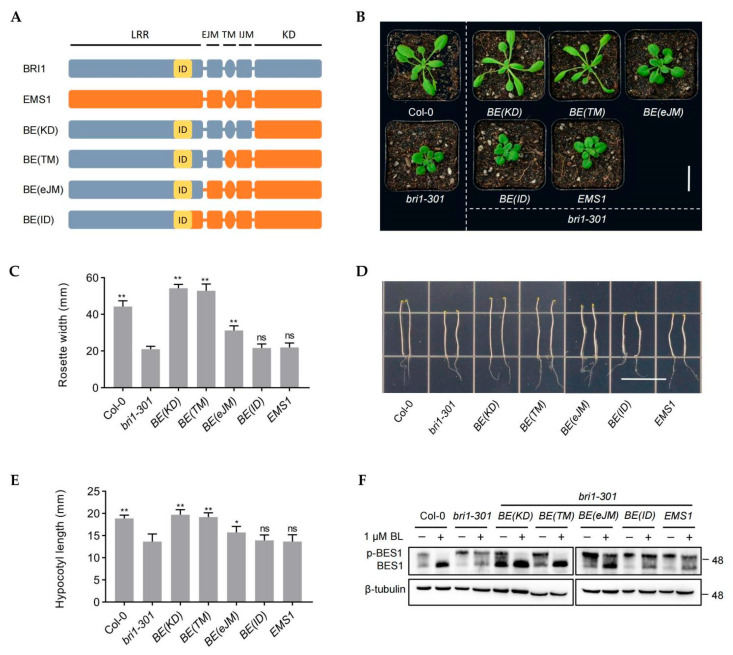
BE(KD), BE(TM) and BE(eJM) could activate BR sigaling. (**A**) Schematic diagram of the extracellular domain of BRI1 fused with different domains of EMS1. The BRI1 and EMS1 protein structures were labeled in gray and orange, respectively. The ID domain is marker in yellow. LRR, leucine-rich repeat; ID, island domain; EJM, extracellular juxtamembrane domain; TM, transmembrane domain; IJM, intracellular juxtamembrane domain; KD, kinase domain. (**B**) Phenotypes of 4-week-old transgenic lines expressing *BE(KD)*, *BE(TM)*, *BE(eJM)*, *BE(ID)*, and *EMS1* under the *BRI1* promoter in *bri1-301* background. Scale bar, 2.0 cm. (**C**) Quantification of the transgenic lines with the diameter of the rosette leaves in a whole plant that grew for 4 weeks, n = 10 plants. Statistical analysis was performed to compare the transgenic plants versus *bri1-301*, ** *p* < 0.001 as one-way ANOVA with a Tukey’s test. (**D**–**E**) The hypocotyl elongation of 7-day-old dark-grown transgenic seedlings in 1/2 MS medium. Scale bar, 1.5 cm. Measurements of hypocotyl length are displayed as means ± SD, n = 10 seedlings. The statistical analysis was performed to compare the transgenic plants’ hypocotyl length versus *bri1-301*, * *p* < 0.05, ** *p* < 0.001 (one-way ANOVA with a Tukey’s test). (**F**) Dephosphorylation of BES1 in plants. Phosphorylated BES1 (pBES1) and dephosphorylated BES1 were detected with BES1 antibody in the extracts of 10-day-old transgentic seedlings. β-tubulin served as the loading control. BL, brassinolide.

**Figure 2 ijms-23-02155-f002:**
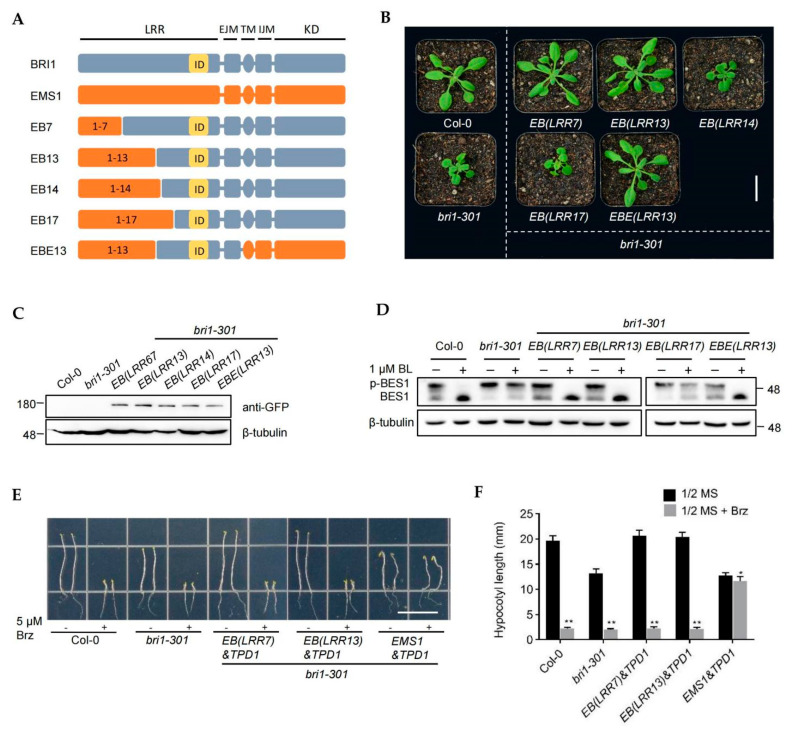
The LRR14-EJM domain is the main differential region of EMS1 and BRI1. (**A**) Schematic diagram of the leucine-rich repeat domain of EMS1 fused with different domains of BRI1. The BRI1 and EMS1 protein structures were labeled in gray and orange, respectively. The ID domain is a marker in yellow. (**B**) Phenotypes of 4-week-old transgenic lines expressing *EB(LRR7)*, *EB(LRR13)*, *EB(LRR14)*, *EB(LRR17)*, and *EBE(LRR13)* under the *BRI1* promoter in *bri1-301* background. Scale bar, 2.0 cm. (**C**) Protein expression levels in the rosette leaves of the corresponding plants shown in (**B**) were detected with anti-GFP antibody. β-tubulin served as the loading control. (**D**) Phosphorylated BES1 (pBES1) and dephosphorylated BES1 were detected with BES1 antibody in the extracts of 10-day-old seedlings. β-tubulin served as the loading control. BL, brassinolide. (**E**) Hypocotyl elongation of *EB(LRR7)&TPD1*, *EB(LRR13)&TPD1*, and *EMS1&TPD1* transgenic seedlings treated with or without 5 μM Brz. Scale bar, 1.5 cm. (**F**) Measurements of hypocotyl length are displayed as means ± SD (n = 10). The statistical analysis was performed to compare the untreated versus treated with Brz samples, * *p* < 0.05, ** *p* < 0.001 (two-way ANOVA with a Sidak’s multiple comparison test).

**Figure 3 ijms-23-02155-f003:**
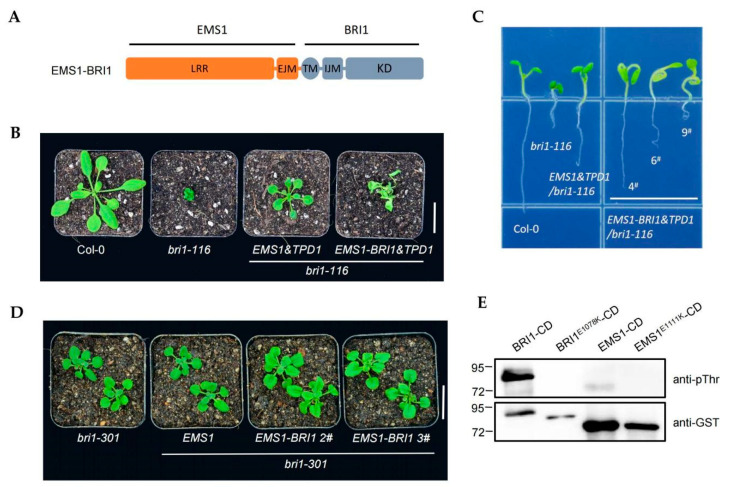
Comparing the kinase activity of EMS1 and BRI1. (**A**) Schematic diagram of chimeric receptor EMS1-BRI1. The extracellular domains of EMS1 fused with the intracellular domains of BRI1. BRI1 and EMS1 protein structures were labeled in gray and orange, respectively. (**B**) Phenotypes of 4-week-old transgenic lines expressing *EMS1&TPD1* and *EMS1-BRI1&TPD1* in *bri1-116* background. Scale bar, 2.0 cm. (**C**) 7-day-old seedlings grown on 1/2 MS medium. (**D**) Phenotypes of 4-week-old *bri1-301* mutants and transgenic lines expressing *EMS1* and *EMS1-BRI1* in *bri1-301* background. Scale bar, 2.0 cm. (**E**) In vitro assay of the kinase activity of recombinant proteins BRI1-CD, BRI1E1078K-CD, EMS1-CD, and EMS1E1111K-CD. Phosphorylation changes were analyzed by pThr antibody. GST served as the loading control.

**Figure 4 ijms-23-02155-f004:**
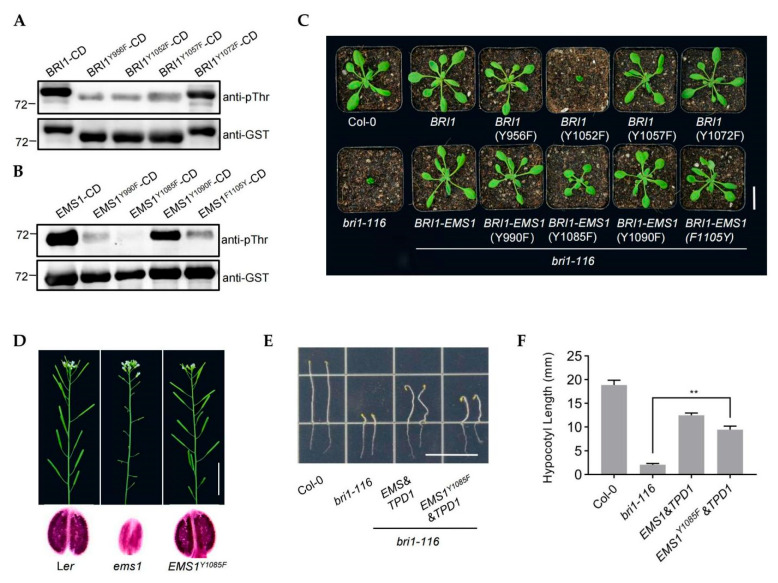
Effects of mutations Tyr residues on EMS1 and BRI1. (**A**) In vitro assay of the kinase activity of recombinant proteins BRI1-CD, BRI1^Y956F^-CD, BRI1^Y1052F^-CD, BRI1^Y1057F^-CD, and BRI1^Y1072F^-CD. Phosphorylation levels were analyzed by pThr/Tyr antibody. GST served as the loading control. (**B**) In vitro assay of the kinase activity of EMS1-CD, EMS1^Y990F^-CD, EMS1^Y1085F^-CD, EMS1^Y1090F^-CD, and EMS1^F1105Y^-CD. Phosphorylation levels were detected by pThr antibody. GST as the loading control. (**C**) Phenotypes of 4-week-old transgenic plants with single-site mutation in *bri1-116* mutant, Scale bar, 2.0 cm. (**D**) Primary inflorescences (**top**) and Alexander staining of pollen grains in mature anthers (bottom) showing the fertility phenotype of *EMS1*^Y1085F^ transgenic plant in *ems1* background. (**E**,**F**) Hypocotyl phenotype of *EMS1&TPD1*/*bri1-116* and *EMS1^Y1085F^&TPD1*/*bri1-116* transgenic seedlings after grown for 7 days in the dark. Scale bar, 1.5 cm. Measurements of hypocotyl length are displayed as means ± SD, n = 10 seedlings. ** *p* < 0.001 (two-tailed *t*-test).

**Figure 5 ijms-23-02155-f005:**
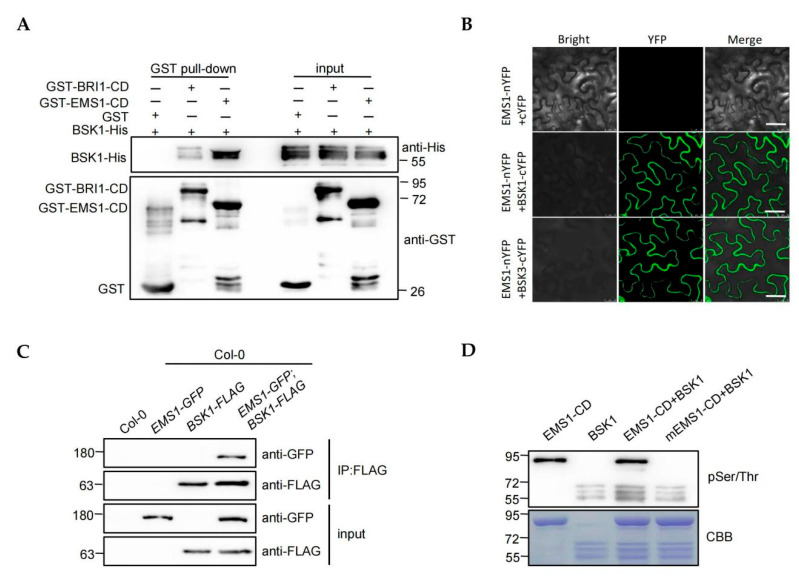
BSKs are the substrate of BRI1 and EMS1. (**A**) Pull-down assays showed BRI1 and EMS1 could interact with BSK1, respectively. GST, GST-BRI1-CD, and GST-EMS1-CD were immobilized on GST beads, incubated with BSK1-His protein, and immunoblot analysis is detected with His antibody and GST antibody. (**B**) BiFC assays showed the interaction of EMS1 with BSK1 and BSK3. The EMS1-nYFP was coexpressed with cYFP, BSK1-cYFP, and BSK3-cYFP in *N. benthamiana*, respectively, after 2 days being visualized under confocal microscopy. cYFP and nYFP represent the C-terminus and N-terminus of YFP, respectively. The EMS1-nYFP+cYFP is the negative control, Scale bar, 50 μm. (**C**) EMS1 interacts with BSK1 in vivo. Total protein was extracted from 1-week-old Col-0, *EMS1-GFP*/Col-0, *BSK1-FLAG*/Col-0, and *EMS1-GFP*; *BSK1-FLAG*/Col-0 seedlings, immunoprecipitated with anti-FLAG M2 affinitygel, and then analyzed by GFP antibody and FLAG antibody. (**D**) EMS1 kinase domain phosphorylates BSK1 in vitro. The kinase assays were performed using EMS1-CD-MBP, mEMS1-CD-MBP, and BSK1-His. Top panel, Phosphorylation analyzed by pSer/Thr. Bottom panel, the gel stained with CBB.

**Figure 6 ijms-23-02155-f006:**
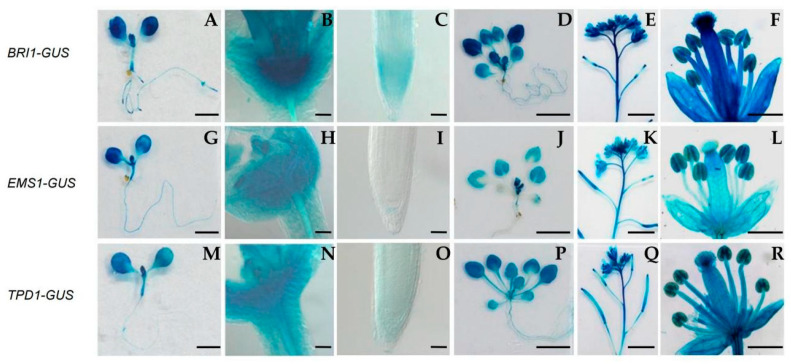
The expression of *BRI1*, *EMS1*, and *TPD1*. (**A**–**F**) *BRI1::GUS*; (**G**–**L**) *EMS1::GUS*; (**M**–**R**) *TPD1::GUS*. From left to right, each small panel represents a seedling 8 days after germination (DAG), a shoot tip from an 8-DAG seedling, a root tip from an 8-DAG seedling, a seedling of 25-DAG, an inflorescence from a 45-DAG plant, and a mature flower from a 45-DAG plant, respectively. Scale bars: (**A**,**G**,**M**) 1 mm; (**B**,**H**,**N**) 50 μm; (**C**,**I**,**O**) 50 μm; (**D**,**J**,**P**) 0.5 cm; (**E**,**K**,**Q**) 0.5 cm; (**F**,**L**,**R**) 1 mm.

**Figure 7 ijms-23-02155-f007:**
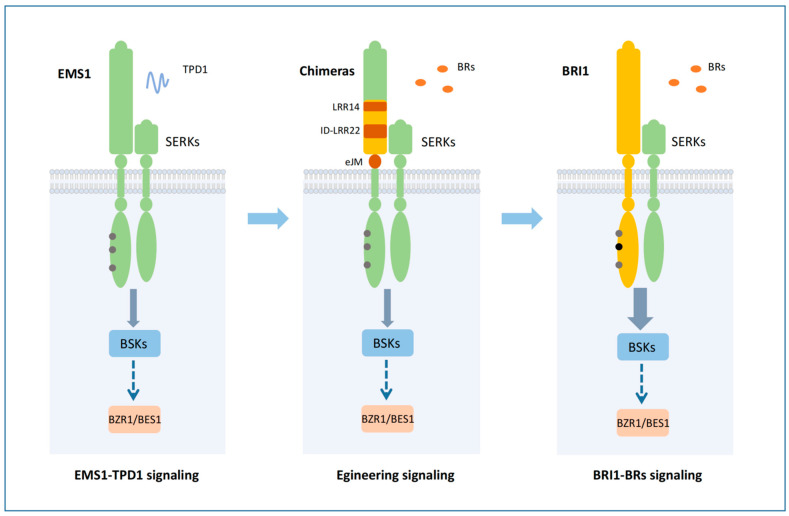
A proposed model for the molecular mechanisms of EMS1 and BRI1 signaling and the origin and divergence of BR receptor. The left side of the model diagram shows the molecular mechanism of EMS1-TPD1 signaling. The ligand TPD1 binds to EMS1 and activates the complex composed of EMS1-SERKs. The intracellular domain of EMS1 interacts with the direct substrate BSKs to initiate the signal transduction pathway. The three tyrosine phosphorylation sites Y990, Y1085, and Y1090 shown in the kinase domain of EMS1 are involved in the regulation of kinase activity. The middle diagram represents the chimeric receptor signaling mechanism. When the LRR14-eJM region in EMS1 was replaced with that of BRI1, the EMS1 chimeric receptor gained the ability as a BR receptor The LRR14-eJM region includes at least three important motifs including LRR14, ID-LRR12, and eJM domain. The right side of the model diagram shows the molecular mechanism of BRI1-BRs signaling. The corresponding phosphorylation sites Y956, Y1052, and Y1057 in the kinase domain of BRI1 are involved in the regulation of kinase activity, but only Y1052 is an essential phosphorylation site for BRI1 function. Due to the stronger kinase activity of BRI1 than EMS1, BRI1 has a stronger signal output ability than EMS1. Our experiments revealed that the functional divergence of BRI1 and EMS1 exists in the receptor’s extracellular domain. Therefore, three important motifs in the LRR14-eJM region may be the key molecular components in the origin of the BR receptor.

## Data Availability

The data or material of this study are available from the corresponding author, B.W.Z., upon reasonable request.

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
