# Peer review of "Engineering Chimeras by Fusing Plant Receptor-like Kinase EMS1 and BRI1 Reveals the Two Receptors’ Structural Specificity and Molecular Mechanisms"

_ijms, 2022, doi:10.3390/ijms23042155_

Round 1

Reviewer 1 Report

  1. Title: "Engineering chimeras by fusing EMS1 and BRI1 2 reveals the specificity of the two receptors" is not informative based on the study. engineering chimeras of what? Please use full words instead of abbreviations.
  2. abstract: There are too many abbreviations. Please use full names. The sentence structure for example, "The receptor-like kinase (RLK) BRI1 perceives BRs to activate a well-known transduction pathway to regulate expression of various genes, and the RLK EMS1 governs tapetum formation via a signaling pathway shared with BRI1." does not make any sense and is desultory in nature. Please formulate sentences with  precision and clarity. There are also too many abbreviations for example, "we identified that the TM, IJM, kinase, and LRR1-13 domains of EMS1." Also, "We compared the kinase domains of EMS1 and BRI1, and found EMS1’s
    19 was weaker than BRI1’s. Importantly, the analysis of phosphorylation sites uncovered that the Y1057 in the kinase domain was essential for the BRI1 function, but the corresponding site in EMS1 was not essential for the function of EMS1, suggesting a site regulation difference in the
    two receptors." is not clear.
  3.  Introduction: The research question should be defined at the beginning of the Intro. Perhaps, "BRI1 is a well-studied LRR-RLK that functions in regulating seed germination, skotomorphogenesis, flowering, fertility, leaf senescence, vascular differentiation, and stress tolerance [3, 21, 22]." and define the problems. Also, why use capital letters such as "EXCESS MICROSPOROCYTES1" and more.
  4. Results: "2.1. TM, IJM and kinase domains, but not the EJM domain, of EMS1 can completely replace the corresponding BRI1 domain" subtitle should use full names instead of abbreviations. Also, "However, it
    remained unclear whether the subdomain outside of the EMS1 kinase domain could replace the corresponding BRI1 domain." why is this important to investigate in plant physiology? Also, "We transformed these chimeric receptors into a weak BRI1 mutant bri1-301." Here, the study used chimeric technology but is not explained in the title and abstract.
  5. "2.2. The LRR14-EJM domain confers the functional divergence between BRI1 and EMS1" functional divergence to do what? Please be precise.
  6. Discussion: The research results or findings should be articulated at the beginning of the discussion. Why are they important to our understanding of plant biology? 
  7. Methods and Materials: "Arabidopsis thaliana Columbia (Col-0) and Landsberg erecta (Ler) ecotypes" plants were used in the study. Please indicate them in the title and abstract. Also, "Constructs and transgenic plants generation" requires more detail on what was used and done.
  8. BiFC assay: Protein-protein interactions were performed using bimolecular fluorescence complementation (BiFC) in N. benthamiana." The sutitle should use full name. Also, another plant was use here. The M&M should have a section on the plants used in the study and the conditions used to grow them.
  9. Coimmunoprecipitation: "Total Proteins were incubated with anti-Flag M2 affinity gel (sigma, A2220)." Should include that anti-FLAG M2 Affinity gel is a mouse monoclonal antibody that is covalently attached to agarose. The antibody binds FLAG at the N-terminal, Met-N-terminal, C-terminal and internal locations of fusion proteins. Binding is calcium-independent.
  10. Data on IP experiments: The data are missing no IP antibody control.
  11. Figure graphs: The stats legend should indicate what is being compared. 
  12. Figure blots: It appears that the blots were manipulated when compared to the ones on the supplementary section. The blot should be the same background. Why not use the full length blot in the figures of the manuscript? Also, the blots should indicate the MWs. 
  13. Fig 1F blot should be quantified using density of pixels of the bands and expressed as a ratio of protein expression to tubulin. The same analyses should be done with the other blots.
  14. Fig 3F: why are the bands of anti-GST different?
  15. Fig 5A: blot of anti-GST requires proper labelling.  Also, Fig 5C is missing no IP antibody control. Also, define "input" on the blot.
  16. The manuscript reveals over 250 grammar errors which require attention. There are too many abbreviations. Also, there is 8% plagiarism which is unacceptable for publication and requires attention.

Author Response

Response to reviewer 1:

  1. Title: "Engineering chimeras by fusing EMS1 and BRI1 reveals the specificity of the two receptors" is not informative based on the study. engineering chimeras of what? Please use full words instead of abbreviations.

Response: We appreciate the reviewer for comments and suggestions. "Engineering chimeras" refers to artificially constructed chimeric receptors, and we multiple engineered chimeric receptors by fusing two receptor genes, BRI1 and EMS1. The abbreviations in the title include EMS1 and BRI1 gene names, the full names are BRI1 (BRASSINOSTEROID INSENSITIVE 1) and (EXCESS MICROICROSPOROCYTES 1), respectively. If the full name is written, the title would be too long. These two genes have been discovered very early, and their names are often used directly in the title. For example, the following paper in this field:

Title: BRI1 controls vascular cell fate in the Arabidopsis root through RLP44 and phytosulfokine  g. Proc. Natl Acad. Sci. USA. 2018.

Title : SYP22 and VAMP727 regulate BRI1 plasma membrane targeting to control plant growth in Arabidopsis. New Phytologist.2019.

Title: BES1 is activated by EMS1-TPD1-SERK1/2-mediated signaling to control tapetum development in Arabidopsis thaliana. Nature Communications.2019.

  1. abstract: There are too many abbreviations. Please use full names. The sentence structure for example, "The receptor-like kinase (RLK) BRI1 perceives BRs to activate a well-known transduction pathway to regulate expression of various genes, and the RLK EMS1 governs tapetum formation via a signaling pathway shared with BRI1." does not make any sense and is desultory in nature. Please formulate sentences with precision and clarity. There are also too many abbreviations for example, "we identified that the TM, IJM, kinase, and LRR1-13 domains of EMS1." Also, "We compared the kinase domains of EMS1 and BRI1, and found EMS1’s was weaker than BRI1’s. Importantly, the analysis of phosphorylation sites uncovered that the Y1057 in the kinase domain was essential for the BRI1 function, but the corresponding site in EMS1 was not essential for the function of EMS1, suggesting a site regulation difference in thetwo receptors." is not clear.

Response: Our study focus on two genes: BRI1 and EMS1. In the writing of our paper, we first want to tell readers what BRI1 and EMS1 are, and why we should study these two genes, but not one of them.

We appreciate the reviewer for comments. We have made changes as the reviewer suggested.

The TM, IJM, kinase, and LRR1-13 are the abbreviations of several sub-domains, and we have added the full names.

  1. Introduction: The research question should be defined at the beginning of the Intro. Perhaps, "BRI1 is a well-studied LRR-RLK that functions in regulating seed germination, skotomorphogenesis, flowering, fertility, leaf senescence, vascular differentiation, and stress tolerance [3, 21, 22]." and define the problems. Also, why use capital letters such as "EXCESS MICROSPOROCYTES1" and more。

Response: Following the pattern in plant biology, the Introduction first should introduce background knowledge related to the research so that readers can quickly understand the author's work. both BRI1 and EMS1 are the main focus of our study which belong to receptor kinase family. The receptor kinase family has commonalities in structure and working mode, for example, it can be divided into three structures, extracellular domain, transmembrane domain, intracellular domain. In addition, co-receptors are often required to assist in signal initiation. In the introduction, we first talk about the structural commonality of the receptor kinase family and their underlying mechanisms, so that the reader can understand the RLK’s features, and these features are shared by both BRI1 and EMS1.

The names of plant genes are generally named according to the mutant that was first screened. For example, the comments of EMS1 on the tair.

  1. Results: "2.1. TM, IJM and kinase domains, but not the EJM domain, of EMS1 can completely replace the corresponding BRI1 domain" subtitle should use full names instead of abbreviations. Also, "However, itremained unclear whether the subdomain outside of the EMS1 kinase domain could replace the corresponding BRI1 domain." why is this important to investigate in plant physiology? Also, "We transformed these chimeric receptors into a weak BRI1 mutant bri1-301." Here, the study used chimeric technology but is not explained in the title and abstract.

Response: We have added the full name instead of abbreviations in the subtitle. For some recurring names, the abbreviated form can be used after the full-length name is indicated for the first time. We will add a comment when the name appears for the first time, because the use of the full name will cause the title to be too long, which will affect reading and understanding. Our study focuses on the molecular structural features of EMS1 and BRI1, and does not directly relate to plant physiology. For the structural study, the first part of our results emphasize on the intracellular structure, while the intracellular differences of EMS1 and BRI1 were not much different. It shows that the functional differentiation of EMS1 and BRI1 is not in the intracellular region, but it should be in the extracellular region. Therefore, it is very important to study the structure of the intracellular region (mainly the extracellular region). Our title Engineering chimeras specifically refers to “To generate chimeric receptors”. And this technique is repeatedly used and mentioned in the study of receptor kinase, and we further discussed it in the discussion section.

  1. "2.2. The LRR14-EJM domain confers the functional divergence between BRI1 and EMS1" functional divergence to do what? Please be precise.

Response: We previously reported that "EMS1 and BRI1 control distinct biological processes through extracellular domain diversity and intracellular domain protection (Zheng et al .2019)", EMS1 and BRI1 are conserved in intracellular, and functional divergence refers to the extracellular domain. Extracellular domain differentiation may involve binding to different ligands, such as EMS1 binding to TPD1, while BRI1 binding to BR, or binding to different co-receptors, EMS1 binding SERK1/2, BRI1 binding SERK1/2/3/4, or affect their signal transmission.

  1. Discussion: The research results or findings should be articulated at the beginning of the discussion. Why are they important to our understanding of plant biology? 

Response:. We agree with the reviewer’s suggestion but as there is no strict format for the discussion to follow, Some authors prefer to summarize the findings in the first paragraph of the discussion, while others summarize the findings at the end of the discussion. Our manuscript does not summarize in the first paragraph of discussion, but at the end of the article. We believe that the Discussion section should be read as a whole. This paper mainly does not directly involve plant physiology or plant biology, but focuses on revealing the specificity of the two receptors EMS1 and BRI1 and the analysis of key segments of the origin of the BR receptor.

  1. Methods and Materials: "Arabidopsis thaliana Columbia (Col-0) and Landsberg erecta (Ler) ecotypes" plants were used in the study. Please indicate them in the title and abstract. Also, "Constructs and transgenic plants generation" requires more detail on what was used and done.

Response: We would like to kindly request the reviewer to let us keep the original title since by including any ecotype characters would lose the original meaning of the study. We described the biological use of Col-0 and Ler in detail in the Methods section.we have added corresponding details on “Constructs and transgenic plants generation” as requested.

  1. BiFC assay: Protein-protein interactions were performed using biomolecular fluorescence complementation (BiFC) in N. benthamiana." The subtitle should use full name. Also, another plant was use here. The M&M should have a section on the plants used in the study and the conditions used to grow them.

Response: We have made changes as the reviewer requested. 

  1. Coimmunoprecipitation: "Total Proteins were incubated with anti-Flag M2 affinity gel (sigma, A2220)." Should include that anti-FLAG M2 Affinity gel is a mouse monoclonal antibody that is covalently attached to agarose. The antibody binds FLAG at the N-terminal, Met-N-terminal, C-terminal and internal locations of fusion proteins. Binding is calcium-independent.

Response: We constructed BSK1-FLAG and BSK3-FLAG vectors with C-terminal fusion of 3×FLAG tag Co-IP experiments. Anti-Flag M2 affinity gel binds FLAG tags at the C-terminus of BSK1 and BSK3. The GFP tag fused to the C-terminus of EMS1, if EMS1 interacts with BSK1 or BSK3, EMS1-GFP can be detected by a GFP antibody in the Co-IP assay. We have added the instruction of antibody the reviewer suggested.

  1. Data on IP experiments: The data are missing no IP antibody control.

Response: Our presented Co-IP assay refers to several relevant literatures (as below). Furthermore, to verify the interaction of EMS1 with BSK1 and BSK3, not only Co-IP but also pull-down and BIFC were used. Various experiments support our conclusion.

Figure 1d[1]  Figure 4B[2]

Figure 1(c)[3]

1.Tang, W.et al. PP2A activates brassinosteroid-responsive gene expression and plant growth by dephosphorylating BZR1. Nat Cell Biol 2011, 13, (2), 124-31.

2.Li, Z.et al. Two SERK receptor-like kinases interact with the receptor-like kinase EMS1 in Anther Cell Fate Determination. Plant physiology 2016, 173.

3.Chen, L.et al. RNA polymerase II associated proteins regulate stomatal development through direct interaction with stomatal transcription factors in Arabidopsis thaliana. New Phytologist 2020, 230.

  1. Figure graphs: The stats legend should indicate what is being compared. 

Response: We have revised the figures legend (Figure 1, Figure 2, Figure S1 and Figure S2) to clarify the data analysis and included indicator lines in the graphs (Figure S4) to point out comparative data.

  1. Figure blots: It appears that the blots were manipulated when compared to the ones on the supplementary section. The blot should be the same background. Why not use the full length blot in the figures of the manuscript? Also, the blots should indicate the MWs. 

Response: The figure blots in the manuscript were exposed under UV conditions, under which protein markers cannot be displayed. Exposure to white light conditions allows protein markers to be displayed. The blot provided in the supplementary section are the merged pictures of the protein marker and the protein under different exposure condition. Under white light, in addition to markers, some spots can also be displayed, but only target protein bands or non-specific bands can be displayed under UV conditions. So some spots in the supplementary section do not appear in the manuscript. We have indicated the MWs in related pictures.

  1. Fig 1F blot should be quantified using density of pixels of the bands and expressed as a ratio of protein expression to tubulin. The same analyses should be done with the other blots.

Response: In this article, the expression of the protein is mainly to show that the target gene is normally expressed in the transgenic plants. Our genetic material and molecular phenotypes are qualitative rather than quantitative. Therefore, our protein pictures in the manuscript has already met this demand.

  1. Fig 3F: why are the bands of anti-GST different?

Response: The phosphorylation level of the protein changes after the site-directed mutagenesis. Protein mobility is inversely proportional to the protein phosphorylation levels: higher the phosphorylation level , slower the protein mobility  and vice versa that is why the band presented for anti-GST are different.

  1. Fig 5A: blot of anti-GST requires proper labelling. Also, Fig 5C is missing no IP antibody control. Also, define "input" on the blot.

Response: We labeled the proteins according to the pull-down labeling method. We have added protein MW in the figures to make the results more detailed. For Fig 5C is response in Question 10. 

  1. The manuscript reveals over 250 grammar errors which require attention. There are too many abbreviations. Also, there is 8% plagiarism which is unacceptable for publication and requires attention.

Response: We checked grammatical errors in the paper as much as possible, and made the shorthand notes as clear as possible, and we removed potential citations and descriptions that may be repeated with other papers. Thank you again for reviewing and revising our paper.

Reviewer 2 Report

The review on the publication by Bai et al under the title ‘Engineering chimeras by using EMS1 and BRI1 reveals the specificity of the two receptors’.

Please revise in the whole manuscript - the name of the genes has to be in Italics.

I missed the main conclusion part in the manuscript.

Figure 2, 5 and 6 add scale bars

I would like authors prepare general scheme demonstrating the molecular mechanism of BRI1 and EMS1. Moreover, authors have to show on the scheme the origin and divergence of BR receptors. Such a scheme will help to understand the main findings and mechanisms.

Author Response

Response to reviewer 2:

1.Please revise in the whole manuscript - the name of the genes has to be in Italics.

Response: We thank the reviewer for reviewing and revising our paper. We have gone through the entire document and revised the name of the genes in italics as requested.

2.I missed the main conclusion part in the manuscript.

Response: As mentioned in the instructions to Authors, the conclusion section is not mandatory. Our finding of this article is fully stated in the Discussion section, so the main conclusion section is not supplied in the manuscript.

3.Figure 2, 5 and 6 add scale bars

Response: We apologize for the careless mistake. We have now added the scales bars and figure legend in the Figure 2, 5 and 6 as the reviewer suggested. 

4.I would like authors prepare general scheme demonstrating the molecular mechanism of BRI1 and EMS1. Moreover, authors have to show on the scheme the origin and divergence of BR receptors. Such a scheme will help to understand the main findings and mechanisms.

Response: We really appreciate this suggestion. We propose a model to summarize our findings so that readers can quickly understand our main findings and mechanisms. Thank you again for reviewing and revising our paper.

Round 2

Reviewer 1 Report

  1. Title: Please consider to change the title. For example, "Engineering chimeras by fusing plant receptor-like kinase EMS1 and Brassinosteroid insensitive 1-associated kinase 1 BRI1 reveals the two receptors structural specificity and molecular mechanisms"
  2. The revised manuscript has two similar figure data for Fig 1 and 2. Please correct.
  3. Abstract: "We compared the kinase domains of EMS1 and BRI1, and found EMS1’s kinase activity was weaker than BRI1’s." Please quantify weaker. Also, "Our study provides insight into the structural specificity and molecular mechanism of BRI1 and EMS1 as well as the origin and divergence of BR receptors." Does this play an important role in the developmental processes of plant grain filling and leaf cell including the plant bulliform cells? 
  4. The white background in the blots indicate manipulation. Why not use the origin full length blots in the supplemental? 

Author Response

  1. Title: Please consider to change the title. For example, "Engineering chimeras by fusing plant receptor-like kinase EMS1 and Brassinosteroid insensitive 1-associated kinase 1 BRI1 reveals the two receptors structural specificity and molecular mechanisms"

Response: We thank the reviewer’s good suggestion. We change the title as suggested “Engineering chimeras by fusing plant receptor-like kinase EMS1 and BRI1 reveals the two receptors structural specificity and molecular mechanisms”. This title is more precise. We just deleted the full name of BRI1 because both EMS1 and BRI1 belong to receptor like kinase, if we write BRI1 full name, we may also need the full name of EMS1, which makes the title too long. Using receptor like kinase to generalize EMS1 and BRI1 is accurate enough.

  1. The revised manuscript has two similar figure data for Fig 1 and 2. Please correct.

Response: We have carefully checked Figure 1 and Figure 2. In this study, the genetic result usually have two states in plant phenotype, recovery or non-recovery, and there is basically no difference in the phenotype of recovery plants, and there is no difference in the phenotype of non-recovery plants, so they may be similar, but it is true.

  1. Abstract: "We compared the kinase domains of EMS1 and BRI1, and found EMS1’s kinase activity was weaker than BRI1’s." Please quantify weaker. Also, "Our study provides insight into the structural specificity and molecular mechanism of BRI1 and EMS1 as well as the origin and divergence of BR receptors." Does this play an important role in the developmental processes of plant grain filling and leaf cell including the plant bulliform cells? 

Response: Our study showed that BRI1 has a stronger kinase activity than EMS1, and the evidence includes: 1) In the comparison of chimeric receptors EMS1 and EMS1-BRI1 as well as EMS1&TPD1 and EMS1-BRI1&TPD, we found that the transgenic plants of EMS1-BRI1 and EMS1-BRI1&TPD1 had more significant phenotypes, which indicated that the kinase activity of BRI1 had stronger signal output. In addition, in in vitro phosphorylation experiments, we found that the kinase activity of EMS1 was significantly lower than that of BRI1, further supporting this conclusion. But it is difficult to quantify in our experiment, one reason is that our current experiment is not sufficient to quantify. And even if we redesign the experiment, it is difficult to achieve the quantitative goal. At present, there are limitations in the plant research field of membrane protein receptor systems. Only expressing the intracellular domain of the purified protein cannot fully reflect the real activity of the nature protein. We point to the stronger kinase activity of BRI1, which may suggest an evolutionary trend in BR receptor activity.

There are many physiological processes regulated by BRI1, including plant grain filling and leaf cell including the plant bulliform cells, and related studies have been reported in Arabidopsis and rice. However, the biological process regulated by EMS1 signaling is relatively specific, and has been reported in Arabidopsis, rice, and maize. Both EMS1 and BRI1 are essential for plant, without these two receptor, plants cannot complete the whole life cycle.

  1. The white background in the blots indicate manipulation. Why not use the origin full length blots in the supplemental? 

Response: We have added the origin full-length blots in the Supplementary file as suggested. Since the protein bands are exposed in UV, the protein marker cannot be displayed. Therefore, in the first submission, the merged image of the original image and the marker was provided, and the original full-length imprint was not provided.  

Thank you again for reviewing and revising our paper.
